FERMILAB-PUB-23-471-CSAID

# Variance Reduction via Simultaneous Importance Sampling and Control Variates Techniques Using Vegas

Prasanth Shyamsundar[1*], Jacob L. Scott[2], Stephen Mrenna[3], Konstantin T. Matchev[4], Kyoungchul Kong[2]

**1** Fermilab Quantum Institute, Fermi National Accelerator Laboratory, Batavia, IL 60510, USA
**2** Department of Physics and Astronomy, University of Kansas, Lawrence, KS 66045, USA
**3** Computational Science and Artificial Intelligence Division, Fermi National Accelerator Laboratory, Batavia, IL 60510, USA
**4** Institute for Fundamental Theory, Physics Department, University of Florida, Gainesville, FL 32611, USA
* prasanth@fnal.gov

January 25, 2024

## Abstract

Monte Carlo (MC) integration is an important calculational technique in the physical sciences. Practical considerations require that the calculations are performed as accurately as possible for a given set of computational resources. To improve the accuracy of MC integration, a number of useful variance reduction algorithms have been developed, including importance sampling and control variates. In this work, we demonstrate how these two methods can be applied simultaneously, thus combining their benefits. We provide a python wrapper, named CoVVVR, which implements our approach in the Vegas program. The improvements are quantified with several benchmark examples from the literature.

# 1 Introduction

In many fields of science, including particle physics, one has to compute the value of an integral

$$F \equiv \int_\Omega d\mathbf{x}\, f(\mathbf{x})\,, \tag{1}$$

over some domain $\Omega$ with volume

$$V(\Omega) \equiv \int_\Omega d\mathbf{x}\,, \tag{2}$$

where $\mathbf{x} \in \mathbb{R}^d$ is a $d$-dimensional vector of independent variables. The function $f(\mathbf{x})$ can be evaluated for a given $\mathbf{x}$, but can be arbitrarily complicated. In high energy physics, integrations like (1) are ubiquitous, arising when computing total cross-sections (or differential cross-sections with respect to low-dimensional event variables), particle lifetimes, convolutions with transfer functions describing detector effects, etc.

Although widespread, this problem is fundamentally challenging — with the exception of a few trivial cases (typically found in the textbooks), the integral cannot be performed analytically, and one has to resort to numerical methods for its evaluation. Monte Carlo (MC) methods are particularly suited for high-dimensional integrals, since their accuracy scales as $\sqrt{\frac{\mathrm{Var}[f]}{N_{\mathrm{trials}}}}$, where $\mathrm{Var}[f]$ is the variance of $f(\mathbf{x})$ over the domain $\Omega$ and $N_{\mathrm{trials}}$ is the number of trials [1]. One obvious way to improve the accuracy is to increase $N_{\mathrm{trials}}$, but this approach soon hits a wall due to resource limitations. Therefore, much attention has been placed on designing variance-reducing techniques. Some of the classic variance reduction methods include importance sampling (IS), stratified sampling, antithetic variables, control variates (CV), etc. [2]. More recent techniques use quasirandom or low-discrepancy pointsets in a class of methods known as Quasi Monte Carlo [3,4], apply multigrid ideas in Multilevel Monte Carlo estimation [5], or leverage machine learning (ML) [6–21]. A parallel research thrust has been the synthesis of ideas, whereby one tries to apply two such techniques, for example, by using several control variates [22], combining antithetic variates and control variates [23, 24], or combining control variates and adaptive importance sampling [25, 26].

In MC methods, random variates are used to sample the function of interest over a number of trials. In importance sampling, one samples points from a distribution $p(\mathbf{x})$ that is, ideally, close to being proportional to $|f(\mathbf{x})|$ ideally. Each sampled point $\mathbf{x}$ as weighted

by the ratio $f(\mathbf{x})/p(\mathbf{x})$, and the integral is estimated as the expected value of the event-weights. While it may be possible to find good sampling distributions analytically, it is typically very difficult in practice. So, one uses techniques like VEGAS [27, 28], Foam [29], and modern neural-network-based techniques [13–21, 30–33] to learn good sampling distributions, which adapt the sampling distribution to better mirror $f$ by computing the value of $f$ at various trial-values. In particular, in the VEGAS algorithm the sampling distribution is adapted by dividing the domain of integration into subregions of equal importance so that the variance of the MC estimate is reduced. A related method, additionally implemented in VEGAS+ [34], is that of stratified sampling, whereby one uses stratification to homogenize the resulting weights within each subregion.

The control variate method, instead, aims to reduce the variance by modifying the integrand. This is accomplished by adding and subtracting a function (the control variate) that is highly-correlated with the integrand and has a known value for its integral. The integral of the control variate converges to the known value, while the variance of the control variate partially cancels the variance in the original integrand. The challenge is to find a control variate that is indeed highly correlated (or anti-correlated) with the integrand $f$. Usually, this can be achieved for only special, low-dimensional cases. Control variates have been used in various domain-inspired applications in [35–41].

The thrust of this paper is to develop a general strategy to combine the well-established importance sampling technique with control variates. We note that the VEGAS method already computes a function that is highly correlated with the integrand in a general way. We illustrate how this function (or its evolution as it converges to an optimal value) can serve as a control variate to improve the accuracy for a given computational budget, or equivalently, to reduce the computation time for a given target accuracy. The paper is accompanied with a python code, CoVVVR, short for CONTROL VARIATES AND VEGAS VARIANCE REDUCTION. The source code is publicly available at `https://github.com/crumpstrr33/covvvr`, together with an introductory tutorial for its usage.

This paper is organized as follows. Section 2 lays the theoretical groundwork of our approach. Sections 2.1, 2.2 and 2.3 briefly review the basic ideas of the Monte Carlo integration, the importance sampling and the control variates methods, respectively. Section 2.4 introduces the joint application of importance sampling and control variates, while Section 2.5 explains how one can leverage the successive VEGAS approximations as control variates. Sec. 3 presents the results from our numerical experiments quantifying the achieved precision improvement on some known benchmark examples. Section 4 contains a description and usage examples of the CoVVVR code. Section 5 is reserved for a summary and conclusions.

## 2 Control Variates and VEGAS

In this section, we describe the main idea of our method using a one-dimensional integration example

$$I = \int_a^b \mathrm{d}x \, f(x) \tag{3}$$

of a real function $f(x)$ from $a$ to $b$, where for definiteness we take $a < b$. In special cases, $a$ can be $-\infty$ and/or $b$ can be $+\infty$.

## 2.1 Naive Monte Carlo Integration

In the naive Monte Carlo approach, $x$ is sampled uniformly over the domain $\Omega$, i.e., $x \sim U(\Omega)$, obtaining $N$ independent and uniformly distributed samples $x_1, x_2, \ldots, x_N$. An estimate $\widehat{I}_N$ of the integral (3) is obtained in terms of the expectation value $E_{x \sim U(\Omega)}[f]$ of the function $f(x)$ over the domain $\Omega$:

$$\widehat{I}_N = V(\Omega) \, E_{x \sim U(\Omega)}\big[f\big] = V(\Omega) \times \frac{1}{N} \sum_{i=1}^{N} f(x_i), \tag{4}$$

where $x$ is sampled uniformly in the domain $\Omega$, i.e., $x_i \sim U(\Omega)$. In the case of a one-dimensional integral as in (3), $V(\Omega) = b - a$.

The uncertainty on $\widehat{I}_N$ can be estimated in terms of the sample variance $\mathrm{Var}\big[\widehat{I}_N\big]$

$$\delta \widehat{I}_N \approx \sqrt{\mathrm{Var}\big[\widehat{I}_N\big]} = V(\Omega) \sqrt{\frac{\mathrm{Var}[f]}{N}}, \tag{5}$$

which decreases as $N^{-1/2}$. However, increasing $N$ arbitrarily is infeasible, as it runs into resource limitations. This motivates methods which attempt to reduce the variance $\mathrm{Var}[f]$ of the integrand function $f(x)$. Two such methods are discussed in the next two subsections.

## 2.2 Importance Sampling

In importance sampling, we choose a sampling function for the random variates that resembles $f(x)$ as closely as possible, and whose integral over the range $(a, b)$ is known. By rescaling with the value of this integral, the new sampling function can be turned into a unit normalized probability distribution function (PDF), which we shall denote with $p(x)$:

$$\int_a^b \mathrm{d}x \, p(x) = 1. \tag{6}$$

The integral of interest (3) can be rewritten as

$$I = \int_a^b \mathrm{d}x \, f(x) = \int_a^b \mathrm{d}x \, p(x) \frac{f(x)}{p(x)} = E_{x \sim p(x)}\left[\frac{f}{p}\right], \tag{7}$$

which can in turn be estimated with Monte Carlo in $N$ trials as

$$\widehat{I}_N = \frac{1}{N} \sum_{i=1}^{N} \frac{f(x_i)}{p(x_i)}, \qquad \text{where } x_i \sim p(x). \tag{8}$$

In analogy to (5) the error on the estimate (8) is given by

$$\delta \widehat{I}_N \approx \sqrt{\mathrm{Var}\big[\hat{I}_N\big]} = \frac{1}{\sqrt{N}} \left[ -I^2 + \int_a^b \mathrm{d}x \, p(x) \frac{f^2(x)}{p^2(x)} \right]^{1/2}. \tag{9}$$

This is minimized when

$$p(x) \propto \big|f(x)\big|. \tag{10}$$

In other words, whenever the two functions $f(x)$ and $p(x)$ have similar shapes, the variance is reduced and the precision of the estimate is improved. In fact, if we could choose $p(x)$ to be exactly proportional to $f(x)$ everywhere, then $\widehat{I}_N$ is exactly equal to $I$ for any $N$. Thus, importance sampling is most beneficial when the function $p(x)$ mimics $f(x)$ as closely as possible.

## 2.3 Control Variates

In contrast to the importance sampling method, which involves a rescaling of the integrand, the control variates method *adds* to $f(x)$ a term The modified integrand can be taken as

$$f_c(x) \equiv f(x) + c\left(g(x) - \frac{G}{b-a}\right), \tag{11}$$

with

$$G \equiv \int_a^b \mathrm{d}x\, g(x) \tag{12}$$

and where $c$ is a parameter which at this point we are free to choose. It is easy to see that the modification (11) does not change the value of the integral, i.e.

$$\int_a^b \mathrm{d}x\, f(x) = \int_a^b \mathrm{d}x\, f_c(x), \qquad \forall c \in \mathbb{R}. \tag{13}$$

$$\widehat{I}_N = V(\Omega)\frac{1}{N}\sum_{i=1}^N f_c(x_i), \qquad \text{where } x_i \sim U(\Omega). \tag{14}$$

We can now leverage the freedom to choose the value of the parameter $c$ in order to minimize the variance. Requiring

$$\frac{\partial \mathrm{Var}[f_c]}{\partial c} = 0 \tag{15}$$

gives the optimal value of $c$ as

$$c^* = -\frac{\mathrm{Cov}(f,g)}{\mathrm{Var}[g]}. \tag{16}$$

The resulting variance is

$$\mathrm{Var}[f_{c^*}] = \mathrm{Var}[f] - \frac{\mathrm{Cov}(f,g)^2}{\mathrm{Var}[g]^2}\mathrm{Var}[g] = \left[1 - \rho^2(f,g)\right]\mathrm{Var}[f], \tag{17}$$

where

$$\rho(f,g) \equiv \frac{\mathrm{Cov}(f,g)}{\sqrt{\mathrm{Var}[f]\mathrm{Var}[g]}} \tag{18}$$

is the familiar Pearson correlation coefficient. Note that if $|\rho(f,g)| > 0$, the variance is necessarily reduced. Furthermore, the higher the correlation between $g(x)$ and $f(x)$, the larger the benefit. Therefore, just like in the method of importance sampling, we desire a function $g(x)$ that i) is highly correlated with $f(x)$ and, ii) has a known expectation value $\mathrm{E}[g] = G/(b-a)$.

The method can be easily generalized to the case of multiple control variates. Appendix A contains the derivation for finding the optimal values $c_i^*$ of the respective coefficients $c_i$ in that case.

### 2.4 Combining Importance Sampling and Control Variates

The two methods discussed in the previous two subsections 2.2 and 2.3 (importance sampling and control variates) can be combined together as follows. Given a known PDF $p(x)$ and a control variate function $g(x)$, we modify the integrand as

$$
I = \int_a^b \mathrm{d}x \; p(x) \left[ \frac{f(x)}{p(x)} + c \left( \frac{g(x)}{p(x)} - E_p \left[ \frac{g}{p} \right] \right) \right]. \tag{19}
$$

The corresponding Monte Carlo estimate is

$$
\widehat{I}_N = \frac{1}{N} \sum_{i=1}^N \left[ \frac{f(x_i)}{p(x_i)} + c \left( \frac{g(x_i)}{p(x_i)} - E_p \left[ \frac{g}{p} \right] \right) \right], \qquad \text{where } x_i \sim p(x). \tag{20}
$$

We would still need a $p(x)$ that is approximately proportional to $f(x)$ and a $g(x)$ whose integral is known, so that $f/p$ is correlated with $g/p$. In this case, the optimal value $c^*$ is given by

$$
c^* = -\frac{\mathrm{Cov}(f/p, g/p)}{\mathrm{Var}[g/p]}. \tag{21}
$$

### 2.5 Repurposing VEGAS Intermediate Outputs as Control Variates

VEGAS is an adaptive and iterative Monte Carlo where, at each iteration $i$, a unit-normalized probability distribution $p_i(x)$ is updated to serve as a probability distribution for importance sampling as described in Section 2.2 [27]. Since VEGAS stores and reports these sampling distributions for each iteration, they can be usefully repurposed as control variates. We can then apply the result in (20), by choosing the IS function as $p(x) = p_n(x)$ for the final iteration $n$ and the control variate function $g(x) = p_i(x)$ from some previous iteration $i < n$. Note that since $p_i(x)$ is unit-normalized the last term in (20) becomes

$$
E_p \left[ \frac{g}{p} \right] = E_{p_n} \left[ \frac{p_i}{p_n} \right] = \int_a^b \mathrm{d}x \; p_n(x) \frac{p_i(x)}{p_n(x)} = \int_a^b \mathrm{d}x \; p_i(x) = 1. \tag{22}
$$

In this way, VEGAS can be used to simultaneously provide both the sampling distribution $p(x)$ and the control variate $g(x)$. In principle, different prescriptions for selecting the previous iteration $i$ to be used in (22) can be designed. Choosing the optimal among those prescriptions can be done on a case by case basis, as discussed below in Section 3.

## 3 Results

### 3.1 Test Cases

For comparison with other studies, we compute values for the benchmark functions presented in Ref. [13]. The domain of integration for each variable is over the range of 0 to 1, and the corresponding values are given in the second column of Table III in Ref. [13]. We confirmed those values as shown in Table 1 and, in what follows, focus on the accuracy of the MC estimate.

For clarity, the definition of the functions used as benchmarks is reproduced below.

### 3.1.1   $d$-dimensional Gaussians

The $d$-dimensional Gaussians are defined as

$$f_1(\mathbf{x}) = \frac{1}{(\sigma\sqrt{\pi})^d} \exp\left(-\frac{1}{\sigma^2}\sum_{i=1}^{d}(x_i - \mu)^2\right), \tag{23}$$

with mean $\mu = 0.5$ and standard deviation $\sigma = 0.2$. They serve as a good starting point and display the effectiveness of control variates for separable functions. We consider $d = 2$, $d = 4$, $d = 8$ and $d = 16$.

### 3.1.2   $d$-dimensional Camel functions

The $d$-dimensional Camel functions are defined by

$$f_2(\mathbf{x}) = \frac{1}{2(\sigma\sqrt{\pi})^d}\left[\exp\left(-\frac{1}{\sigma^2}\sum_{i=1}^{d}(x_i - \mu_1)^2\right) + \exp\left(-\frac{1}{\sigma^2}\sum_{i=1}^{d}(x_i - \mu_2)^2\right)\right] \tag{24}$$

in terms of three parameters, $\mu_1 = 1/3$, $\mu_2 = 2/3$, and $\sigma = 0.2$. Unlike the Gaussians case, the integration variables are not separable. We consider $d = 2$, $d = 4$, $d = 8$ and $d = 16$.

### 3.1.3   Entangled circles

This function is given by

$$f_3(x_1, x_2) = x_2^a \exp\left[-w\left|(x_2 - p_2)^2 + (x_1 - p_1)^2 - r^2\right|\right]$$
$$+ (1 - x_2)^a \exp\left[-w\left|(x_2 - (1 - p_2))^2 + (x_1 - (1 - p_1))^2 - r^2\right|\right] \tag{25}$$

It is largely concentrated on two overlapping circles of radius $r$ with centers at $(p_1, p_2)$ and $(1 - p_1, 1 - p_2)$, respectively. For the numerical experiments, we use $p_1 = 0.4$, $p_2 = 0.6$, $r = 0.25$, $w = 1/0.004$, and $a = 3$.

### 3.1.4   Annulus with cuts

The fourth function is an annulus defined by cuts at $r_{\min}$ and $r_{\max}$:

$$f_4(x_1, x_2) = \begin{cases} 1 & \text{if } r_{\min} < \sqrt{x_1^2 + x_2^2} < r_{\max}, \\ 0 & \text{else.} \end{cases} \tag{26}$$

The cut parameters are chosen to be $r_{\min} = 0.2$ and $r_{\max} = 0.45$.

### 3.1.5   One-loop scalar box integral

The fifth function represents a one-loop scalar box integral encountered in the calculation of $gg \to gh$, an important contribution to Higgs production from gluon fusion. The integral is

$$\begin{aligned} I_5 = \quad & S_{\text{Box}}(s_{12}, s_{23}, s_1, s_2, s_3, s_4, m_t^2, m_t^2, m_t^2, m_t^2) \\ + \quad & S_{\text{Box}}(s_{23}, s_{12}, s_2, s_3, s_4, s_1, m_t^2, m_t^2, m_t^2, m_t^2) \\ + \quad & S_{\text{Box}}(s_{12}, s_{23}, s_3, s_4, s_1, s_2, m_t^2, m_t^2, m_t^2, m_t^2) \\ + \quad & S_{\text{Box}}(s_{23}, s_{12}, s_4, s_1, s_2, s_3, m_t^2, m_t^2, m_t^2, m_t^2), \end{aligned} \tag{27}$$

where

$$S_{\text{Box}}(s_{12}, s_{23}, s_1, s_2, s_3, s_4, m_1^2, m_2^2, m_3^4, m_4^2) = \int_0^1 \frac{\mathrm{d}x_1\, \mathrm{d}x_2\, \mathrm{d}x_3}{\widetilde{\mathcal{F}}_{\text{Box}}^2} \tag{28}$$

and

$$\begin{aligned}
\widetilde{\mathcal{F}}_{\text{Box}} = & - s_{12}x_2 - s_{23}x_1 x_3 - s_1 x_1 - s_2 x_1 x_2 - s_3 x_2 x_3 - s_4 x_3 \\
& + (1 + x_1 + x_2 + x_3)(x_1 m_1^2 + x_2 m_2^2 + x_3 m_3^2 + m_4^2).
\end{aligned} \tag{29}$$

The integrand function $f_5(x_1, x_2, x_3)$ in this case is a sum of four terms of the type $1/\widetilde{\mathcal{F}}_{\text{Box}}^2$. We test with the same parameters as in [13]: $s_{12} = 130^2$, $s_{23} = -s_{12}$, $s_1 = s_2 = s_3 = 0$, $s_4 = 125^2$, and $m_i = 173.9$ for $i = 1 - 4$.

### 3.1.6   Polynomial functions

The final set of integrand functions are quadratic polynomials of $d$ variables

$$f_6(\mathbf{x}) = \sum_{i=1}^{d} x_i(1 - x_i). \tag{30}$$

We consider $d = 18$, $d = 54$ and $d = 96$.

### 3.2   Numerical Comparisons

Results from integrating the six types of test functions from Section 3.1 are shown in Table 1. The first two columns list the name of the function and the dimension of the integration. The third column shows the expected true answer, while the next two columns give the MC estimates from VEGAS and from CoVVVR, respectively. For the latter, we use the one control variate which gives the maximum variance reduction, as explained

| Function | Dim | True Value | Mean | | Normalized RMS Error | |
|---|---|---|---|---|---|---|
| | | | VEGAS | CVInt | VEGAS | CVInt |
| Gaussian | 2 | 0.999186 | 0.999190 | 0.999188 | 1.0049e-04 | 9.1006e-05 |
| | 4 | 0.998373 | 0.998379 | 0.998379 | 0.000145 | 0.000132 |
| | 8 | 0.996749 | 0.996750 | 0.996866 | 0.000202 | 0.000231 |
| | 16 | 0.993509 | 0.993507 | 0.993507 | 0.000293 | 0.000265 |
| Camel | 2 | 0.981660 | 0.981618 | 0.981617 | 0.001286 | 0.001283 |
| | 4 | 0.963657 | 0.963559 | 0.963560 | 0.003180 | 0.003173 |
| | 8 | 0.928635 | 0.929091 | 0.929223 | 0.010283 | 0.010295 |
| | 16 | 0.862363 | 0.784645 | 0.784860 | 0.546183 | 0.544413 |
| Entangled Circles | 2 | 0.013680 | 0.013687 | 0.013687 | 0.004349 | 0.004348 |
| Annulus with Cuts | 2 | 0.127627 | 0.127644 | 0.127642 | 0.003773 | 0.003789 |
| Scalar Top Loop | 3 | 1.9374e-10 | 1.9370e-10 | 1.9370e-10 | 0.000328 | 0.000322 |
| Polynomial | 18 | 3.000000 | 2.999998 | 2.999999 | 2.6498e-05 | 2.1906e-05 |
| | 54 | 9.000000 | 8.999992 | 8.999995 | 4.1208e-05 | 2.9833e-05 |
| | 96 | 16.000000 | 15.999929 | 15.999954 | 7.2611e-05 | 5.1897e-05 |

Table 1: Results for the test case integrals described in Section 3.1 with $n = 50$ iterations, $N_{\text{events}} = 5,000$ events per iteration, and averaged over $N_{\text{runs}} = 1,000$ runs. The results displayed in the CVInt column were obtained with the one control variate that gives maximum variance reduction. The normalized RMS error is given by (31).

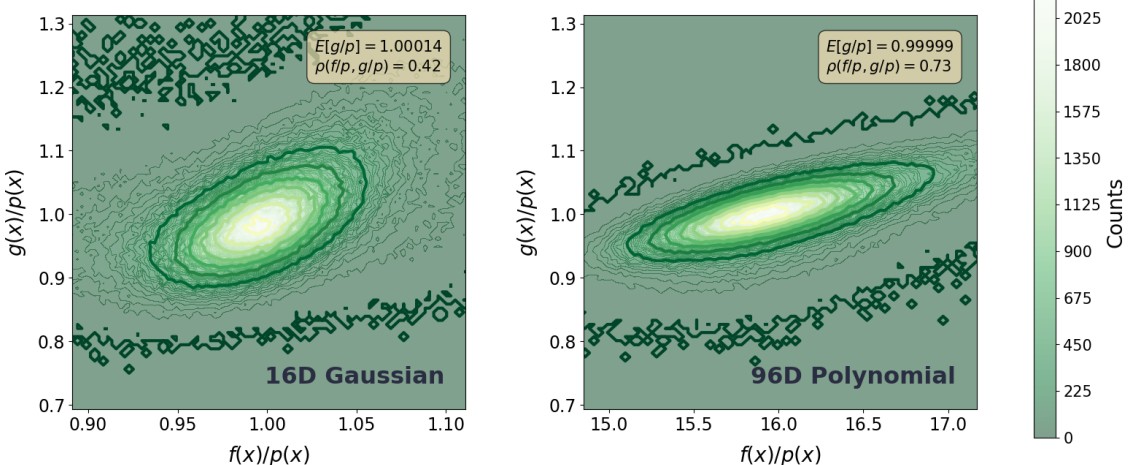

Figure 1: Correlation between the function $f/p$ and the control variate $g/p$. This is shown for the 16d Gaussian (left) and the 96d polynomial (right) run for $n = 100$ iterations with, at most, $N_{\text{events}} = 10000$ evaluations per iteration. The iteration used for the control variate (CV) was chosen automatically using the parameter `cv_iters="auto1"` as described in Section 4. We see that there is a correlation between the functions shown both qualitatively by the plot and quantitatively by a correlation coefficient of $\rho \approx 0.42$ (left) and $\rho \approx 0.75$ (right). Additionally, the expectation is approximately 1, in agreement with (22).

below, and choose the optimal value of $c^*$ according to (21). In the last two columns we show the normalized RMS error defined as

$$\frac{1}{I} \sqrt{\frac{1}{N_{\text{runs}}} \sum_{i=1}^{N_{\text{runs}}} (\widehat{I}_i - I)^2}. \tag{31}$$

We see that, as expected, the accuracy using a CV is comparable or improved.

As implied by eq. (19), the variance reduction results from the presence of a correlation between the function ratios $f(x)/p(x)$ and $g(x)/p(x)$. This correlation is illustrated in Figure 1 for the case of the 16-dimensional Gaussian (left) and the 96-dimensional polynomial (right) function. The iteration used for the CV was chosen automatically. The correlation is readily visible by eye, and the correlation coefficient is $\rho \approx 0.42$ (left) and $\rho \approx 0.75$ (right).

The effect of adding more than one CV is illustrated in Table 2 and Figure 2. We show results with one (1 CV), two (2 CV) or all 49 intermediate approximations (All CVs) from VEGAS as control variates. The results are quoted in terms of the variance reduction in percentage (VRP) and the corresponding computational cost (in seconds, as well as relative to the VEGAS benchmark time). In each case, we select the CV or CVs that reduce the variance the most. Table 2 confirms that, by construction, the use of CVs always improves the accuracy of the estimate. The size of the VRP effect depends on the type of function at hand, and can vary from $\sim 1\%$ to as much as 50% for one CV and 60% for two CVs. The associated computational cost is an increase of about $1.5 - 2.5$ times for one CV and $2 - 4$ times for 2 CVs. Note that an increase in accuracy could also be achieved by increasing the number of events used in a standard VEGAS calculation. We discuss the benefits of the CV method later.

The dependence of the variance reduction on the choice of CVs is illustrated in Figure 2. The heat map in each panel depicts the variance reduction due to 1 CV (along the diagonal)

| | | VEGAS | 1 CV | | 2 CVs | | All CVs | |
|---|---|---|---|---|---|---|---|---|
| Function | Dim | Time (s) | VRP | Time (s) | VRP | Time (s) | VRP | Time (s) |
| Gaussian | 2 | 0.07 | 17.02% | 0.09 (1.4) | 31.40% | 0.14 (2.1) | 47.15% | 1.76 (26.2) |
| | 4 | 0.10 | 15.86% | 0.14 (1.5) | 26.55% | 0.22 (2.3) | 39.78% | 2.91 (30.1) |
| | 8 | 0.15 | 17.28% | 0.25 (1.7) | 23.11% | 0.38 (2.6) | 33.95% | 5.74 (39.4) |
| | 16 | 0.23 | 13.95% | 0.49 (2.1) | 17.22% | 0.79 (3.4) | 23.87% | 13.34 (56.9) |
| Camel | 2 | 0.07 | 0.38% | 0.09 (1.4) | 0.81% | 0.14 (2.0) | 1.24% | 1.75 (25.7) |
| | 4 | 0.10 | 0.12% | 0.14 (1.5) | 0.33% | 0.22 (2.2) | 0.66% | 2.94 (30.0) |
| | 8 | 0.15 | 0.20% | 0.25 (1.6) | 0.25% | 0.39 (2.5) | 0.37% | 5.86 (37.9) |
| | 16 | 0.26 | 0.74% | 0.52 (2.0) | 3.17% | 0.81 (3.2) | 7.72% | 13.70 (53.4) |
| Entangled Circles | 2 | 0.08 | 0.31% | 0.10 (1.2) | 0.36% | 0.15 (1.8) | 0.67% | 1.81 (22.4) |
| Annulus with Cuts | 2 | 0.05 | 1.25% | 0.08 (1.6) | 2.94% | 0.12 (2.3) | 16.93% | 1.77 (33.1) |
| Scalar Top Loop | 3 | 0.12 | 7.31% | 0.14 (1.2) | 49.33% | 0.21 (1.8) | 57.91% | 2.30 (19.9) |
| Polynomial | 18 | 0.26 | 29.50% | 0.56 (2.2) | 29.74% | 0.89 (3.5) | 51.36% | 15.49 (59.9) |
| | 54 | 0.70 | 42.65% | 1.65 (2.4) | 58.97% | 2.63 (3.8) | 71.77% | 47.94 (68.8) |
| | 96 | 1.33 | 49.63% | 3.04 (2.3) | 63.77% | 4.79 (3.6) | 78.18% | 86.55 (65.3) |

Table 2: Results for the variance reduction in percent (VRP) after using one, two, or all 49 intermediate approximations from VEGAS as control variates. In each case, we ran $n = 50$ iterations and at most $N_{\text{events}} = 5000$ events per iteration averaged over $N_{\text{runs}} = 100$ independent runs. The time column shows the corresponding computational cost (in seconds and relative to the VEGAS benchmark time).

and 2 CVs for all combinations. We show results averaged over 10 runs for a 16d Gaussian (left panel) and the scalar top loop (right panel). In each run, we perform 50 iterations with at most 25,000 evaluations per iteration. Along each axis, each panel contains a plot of the normalized variance (red line) and the correlation coefficient between the target function $f(x)$ and the respective CV (purple line).

Figure 2 shows that the optimal iteration for choosing a CV can vary significantly — in the case of a 16d Gaussian, it is around 30, while for the scalar top loop integral, it is at the beginning. When we have the freedom to choose two CVs, some interesting patterns appear as shown in the heat maps, and the optimal choice for the Gaussian are the iterations around 25 and 40, respectively.

Since the optimal choice of the iteration is *a priori* unknown, we show in Table 3 the corresponding results when the iteration is decided and fixed at the very beginning. In this case, we pick the CV from the iteration which is 1/4 of the way from the beginning, i.e., $i = \lfloor n/4 \rfloor$. The number of events $N_{\text{events}}$ and total number of iterations $n$ are chosen based on reaching the precision required in Ref. [13], namely, relative uncertainty of $10^{-4}$ for the first 11 cases and $10^{-5}$ for the last three. We see that even when the CV is fixed rather arbitrarily, the variance reduction is still significant, and can be $\sim 50\%$, as in the case of the polynomial functions.

# 4   CoVVVR: **C**ontrol **V**ariates & **V**egas **V**ariance **R**eduction

The paper is accompanied with a python code, CoVVVR, short for CONTROL VARIATES & VEGAS VARIANCE REDUCTION, that can be used to reproduce our results. The source code is publicly available at `https://github.com/crumpstrr33/covvvr`. In this section, we provide an introduction tutorial.

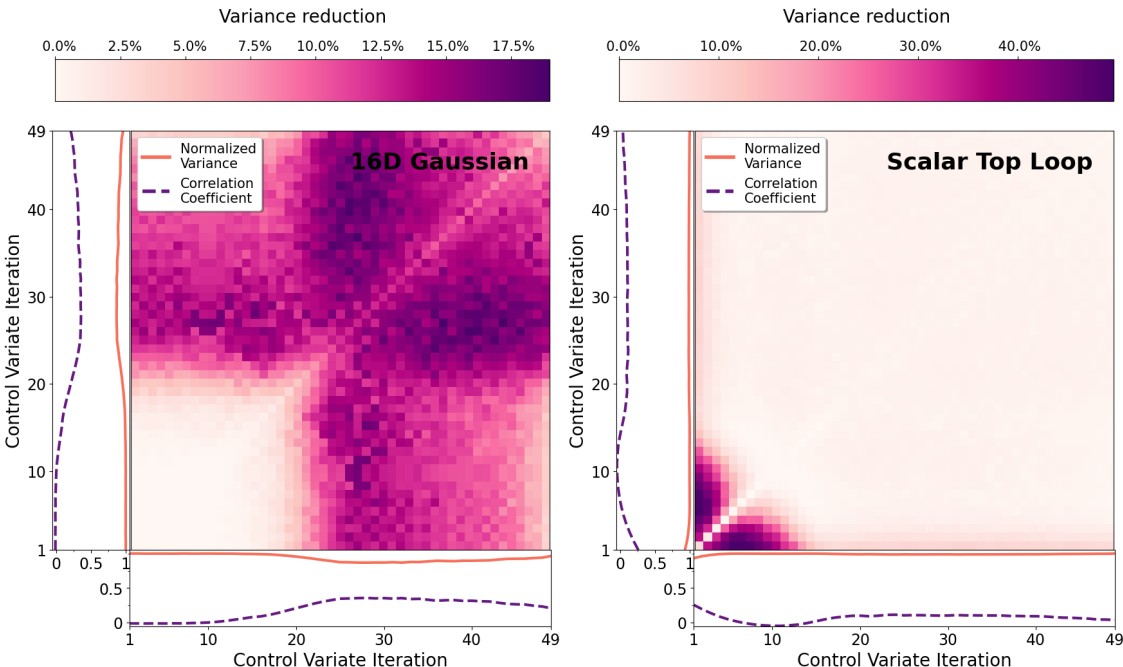

Figure 2: The variance reduction due to 1 control variate (along the diagonal) and 2 control variates for all combinations. We show results for $n = 50$ iterations with at most $N_{\text{events}} = 5{,}000$ evaluations per iteration averaged over $N_{\text{runs}} = 100$ runs for a 16d Gaussian (left panel) and the scalar top loop (right panel). The solid red line shows the normalized variance, i.e., the variance when using the $i$-th control variate iteration divided by the variance obtained without any control variates.

**Installation**: CoVVVR can be installed via `pip`:

```
$ python -m pip install covvvr
```

**Usage**: The workflow involves creating a class that inherits the `Function` subclass and passing that to the `CVIntegrator`. The `Function` class contains the function to be integrated but also other information such as its name, the true value of the integration (if available) and parameters of the function. This class can be a built-in function, such as those used in this paper, or a custom-made function. The `CVIntegrator` class does the integration and stores the results like mean and variance.

**Using a Built-In Function**:

```
1  from covvvr import CVIntegrator
2  from covvvr.functions import NGauss
3
4  # Create 16-dimensional Gaussian
5  ng = NGauss(16)
6
7  # Print out all parameters of the class instance
8  print(ng, '\n')
9
```

| Function | Dim | Events | Total Iter | CV Iter | Standard Deviation | | VRP |
|---|---|---|---|---|---|---|---|
| | | | | | VEGAS | CVInt | |
| Gaussian | 2 | 150,342 | 32 | 8 | 1.256e-04 | 1.246e-04 | 1.58% |
| | 4 | 703,711 | 150 | 37 | 8.288e-05 | 7.976e-05 | 7.37% |
| | 8 | 1,435,484 | 311 | 77 | 8.268e-05 | 7.980e-05 | 6.85% |
| | 16 | 2,775,411 | 614 | 153 | 8.513e-05 | 7.980e-05 | 12.13% |
| Camel | 2 | 342,915 | 73 | 18 | 1.053e-03 | 1.053e-03 | 0.00% |
| | 4 | 19,379,835 | 4,108 | 1,027 | 3.453e-04 | 3.451e-04 | 0.15% |
| | 8 | 47,220,394 | 10,000 | 2,500 | 7.122e-04 | 7.116e-04 | 0.16% |
| | 16 | 1,162,234 | 246 | 61 | 7.860e-02 | 7.860e-02 | 0.00% |
| Entangled Circles | 2 | 48,399,992 | 10,000 | 2,500 | 4.357e-06 | 4.352e-06 | 0.23% |
| Annulus with Cuts | 2 | 1,862,585 | 389 | 97 | 4.308e-04 | 4.292e-04 | 0.74% |
| Scalar-top-loop | 3 | 110,748 | 24 | 6 | 6.436e-14 | 6.433e-14 | 0.12% |
| Polynomial | 18 | 1,807,196 | 400 | 100 | 2.836e-05 | 2.223e-05 | 38.56% |
| | 54 | 4,450,434 | 985 | 246 | 7.299e-05 | 5.346e-05 | 46.37% |
| | 96 | 13,784,476 | 3,051 | 762 | 1.561e-04 | 1.049e-04 | 54.79% |

Table 3: Comparison between native VEGAS and the addition of a single control variate (CV). Number of events and iterations are chosen based on reaching the precision required in Ref. [13], namely, relative uncertainty of $10^{-4}$ for the first 11 cases and $10^{-5}$ for the last three. There are a maximum of $N_{\text{events}} =$5,000 events per iteration. The iteration chosen for the CV is set at a quarter of the total number of iterations. VRP is the variance reduction in percentage; it is the reduction in variance between VEGAS and CVInt by percentage.

```
10    # Create integrator classm & use the 20th iteration as control variate
11    cvi = CVIntegrator(ng, evals=1000, tot_iters=50, cv_iters=20)
12
13    # Run the integration
14    cvi.integrate()
15
16    # Print info
17    cvi.compare(rounding=5)
```

This integrates a 16-dimensional Gaussian with 50 iterations and 5000 evaluations per iteration in VEGAS. It uses the 20th iteration adaptation from VEGAS as the control variate. The output is

```
NGauss(dimension=16, name=16D Gaussian, true_value=0.9935086032227194,
mu=0.5, sigma=0.2)

          |    No CVs   |   With CVs
---------+-------------+-------------
Mean     |     0.99528 |     0.99579
Variance | 4.17808e-06 | 3.54189e-06
St Dev   |     0.00204 |     0.00188
VRP      |             |   15.22696%
```

**Adding User-Defined Function**: The `make_func` function allows for the definition of a user-defined functions via the `Function` subclass. As an example, consider the 2-dimensional function $f(x_1, x_2) = ax_1^2 + bx_2^2$. It can be defined as

```python
1   from covvvr import CVIntegrator, make_func
2
3   # Create function, note that it is vetorized using Numpy slicing
4   def f(self, x):
5       return self.a * x[:, 0]**2 + self.b * x[:, 1]
6
7   # Creating class with name 'WeightedPoly' and assigning values to the
8   # parameters in the function
9   wpoly = make_func(
10      cname='WeightedPoly',
11      dimension=2,
12      function=f,
13      name='Weighted Polynomial',
14      a=0.3,
15      b=0.6
16  )
17
18  # Print out parameters of class (note `true_value` isn't shown but it
19  # can be added if one wants to keep track of that
20  print(wpoly, '\n')
21
22  # Create integrator class and use multiple control variates
23  cvi = CVIntegrator(
24      function=wpoly,
25      evals=1000,
26      tot_iters=50,
27      cv_iters=[10, 15, 20, 25, 30, 35]
28  )
29
30  # Run the integration
31  cvi.integrate()
32
33  # Print info
34  cvi.compare(rounding=5)
```

which outputs

```
WeightedPoly(dimension=2, name=Weighted Polynomial, a=0.3, b=0.6)

         |    No CVs    |   With CVs
---------+-------------+-------------
Mean     |     0.39984 |     0.40004
Variance | 5.83781e-08 | 4.90241e-08
St Dev   | 2.41616e-04 | 2.21414e-04
VRP      |             |   16.02309%
```

Note that vectorization of the integrand greatly increases the speed of the computation.

To not have to deal with classes, one can use the `classic_integrate` function that does the steps above and returns the `CVIntegrator` object. To run the previous code block, one would use

```
1   cvi = classic_integrate(
2       function=f,
3       evals=1000,
4       tot_iters=50,
5       bounds=[(0, 1), (0, 1)],
6       cv_iters=20,
7       cname="WeightPoly",
8       name="Weighted Polynomial",
9       a=0.3,
10      b=0.6
11  )
12  cvi.compare()
```

which outputs the same results as before.

```
         |   No CVs   |  With CVs
---------+-----------+-----------
Mean     |     0.400 |     0.400
Variance | 5.826e-08 | 5.025e-08
St Dev   | 2.414e-04 | 2.242e-04
VRP      |           |   13.746%
```

Note that in the `classic_integrate` function, `bounds` is not optional as it is needed in order to define the dimension of the integral. (In contrast, the `bounds` argument is optional for the `CVIntegrator` class, since `CVIntegrator` determines the dimension of the integral from the `Function` class being passed, and then sets the limits of the integration from 0 to 1 by default.)

**Specifying Control Variate Iterations**: There are multiple valid arguments that can be passed to `cv_iters` for both the `CVIntegrator` class and `classic_integrate` which we list here.

- For using one control variate, pass an integer representing the iteration to use.

- For multiple control variates, pass a list of integers representing the iterations to use.

- The string 'all' will use every iteration.

- The string `all%n` will use every iteration mod $n$. So if one specifies `tot_iters=15` and `cv_iters='all%3'`, then then the iterations used will be [3, 6, 9, 12]

- The previous result can be shifted by using `all%n+b` where $b$ is the shift. So for `tot_iters=15` and `cv_iters='all%3+2'`, you'll get [2, 5, 8, 11, 14].

- The string `auto1` will estimate the best single CV to use by sampling each possibility using `auto1_neval` samples specified by the user.

**Manual Use of Function Class**: To access the function call manually, use `function` or `f`. Using the second example, one can run

```
1   wpoly.f([1.2, 1.5])                    # returns 1.3319999999999999
2   wpoly.f([0.2, 1], [0.8, 0.8], [1, 2])  # returns array([0.612, 0.672, 1.5])
```

This wraps around the private, vectorized `_function`/`_f` used by VEGAS.

## 5   Summary and Outlook

The MC method is widely used in many disciplines, and the issue of variance reduction in MC estimates has existed as long as the method. A number of techniques have been developed to reduce the variance of such MC estimates relative to brute force sampling, including importance sampling and control variates. In this paper, we combined these two methods by leveraging the ability of the importance sampling method implemented in VEGAS to automatically provide viable control variate candidates. We demonstrate and quantify the resulting variance reduction using several benchmark functions considered previously in a similar context [13]. Our new approach was able to reduce the variance by $\mathcal{O}(1\text{-}50)\%$ using 1 CV and $\mathcal{O}(1\text{-}60)\%$ using 2 CVs.

The reduced variance comes at the cost of some computational time. In our experiments, the main source of additional computational overhead in our approach is the computation of the control variates for all the datapoints, using the intermediate VEGAS distributions. The inherent theoretical advantages of our scheme should always be weighed against the possibility to improve the precision by simply increasing the number of data points. The recommended use case of our technique is in situations where the latter approach is comparatively less effective, which will be the case if computing the integrand $f$ is sufficiently computationally expensive. Potential examples of such situations in particle physics include inference involving the slow simulation of the experimental apparatus, or the presence of additional convolutions in the definition of the integrand $f(\mathbf{x})$ itself, e.g., due to transfer functions. Note that the technique proposed here is meant to improve the estimate on the *integral* of $f(\mathbf{x})$, and is not intended for the task of improving event generation. The weights of the events are modified under our technique in such a way that the resulting differential distributions are different from $f(\mathbf{x})$. The contributions of the control variates only cancel out after integrating over the full phase space.

There are several possible avenues for refining the technique proposed in this paper. The VEGAS algorithm has a few parameters which control how the sampling distributions are iteratively adapted. The effect of these parameters on the performance of the VEGAS-based control variates can be studied in order to construct good controls. Furthermore, the VEGAS grid adaptation algorithm could potentially be modified for the specific purpose of providing good control variates. There are other possible avenues for reducing the variance of a MC estimate which leverage deep learning [42,43] or specific domain knowledge about

the integrand [44–51]. Ideally, one would also like to better understand for what types of integrand functions $f(\mathbf{x})$ our technique is more likely to produce a significant improvement (see Table 2). These ideas are left for future work.

## Acknowledgements

The idea for this work arose during the Summer 2022 workshop "Interplay between Fundamental Physics and Machine Learning" at the Aspen Center for Physics, which is supported by National Science Foundation grant PHY-1607611. The authors would like to thank the Aspen Center for Physics for hospitality during the summer of 2022.

**Funding information**   This work is supported in parts by US DOE DE-SC0024407 and DOE DE-SC0022148. JS is supported by University of Kansas Research Excellence Initiative Award and US DOE AI-HEP grant DE-SC0024673. SM and PS are partially supported by the U.S. Department of Energy, Office of Science, Office of High Energy Physics QuantISED program under the grants "HEP Machine Learning and Optimization Go Quantum", Award Number 0000240323, and "DOE QuantiSED Consortium QCCFP-QMLQCF", Award Number DE-SC0019219. This manuscript has been authored by Fermi Research Alliance, LLC under Contract No. DEAC02-07CH11359 with the U.S. Department of Energy, Office of Science, Office of High Energy Physics.

## Code and data availability

The code and data that support the findings of this study are openly available at the following URL: `https://github.com/crumpstrr33/covvvr`.

## A   Multiple Control Variates

For $N_{\mathrm{CV}} > 1$ control variates $g_i(\mathbf{x})$, the integrand is modified in analogy to (11)

$$f_c(\mathbf{x}) = f(\mathbf{x}) + \sum_{i=1}^{N_{\mathrm{CV}}} c_i(g_i(\mathbf{x}) - E[g_i]) \tag{32}$$

and its variance is:

$$
\begin{aligned}
\mathrm{Var}[f_c] &= \mathrm{Var}\left[ f(\mathbf{x}) + \sum_{i=1}^{N_{\mathrm{CV}}} c_i(g_i(\mathbf{x}) - E[g_i]) \right] \\
&= \mathrm{Var}[f] + 2\mathrm{Cov}\left( f, \sum_{i=1}^{N_{\mathrm{CV}}} c_i(g_i(\mathbf{x}) - E[g_i]) \right) + \mathrm{Var}\left[ \sum_{i=1}^{N_{\mathrm{CV}}} c_i(g_i(\mathbf{x}) - E[g_i]) \right] \\
&= \mathrm{Var}[f] + 2\mathrm{Cov}\left( f, \sum_{i=1}^{N_{\mathrm{CV}}} c_i g_i \right) + \mathrm{Var}\left[ \sum_{i=1}^{N_{\mathrm{CV}}} c_i g_i \right] \\
&= \mathrm{Var}[f] + 2\sum_{i=1}^{N_{\mathrm{CV}}} c_i \mathrm{Cov}(f, g_i) + \sum_{ij}^{N_{\mathrm{CV}}} c_i c_j \mathrm{Cov}(g_i, g_j)
\end{aligned}
\tag{33}
$$

where $\text{Cov}(g_i, g_i) = \text{Var}[g_i]$. Taking derivatives with respect to the coefficients $c_j$ gives

$$\frac{\partial \text{Var}[f_c]}{\partial c_j} = 2\text{Cov}(f, g_j) + 2\sum_{i=1}^{N_{\text{CV}}} c_i \text{Cov}(g_i, g_j). \tag{34}$$

Setting that equal to zero implies

$$\sum_{i=1}^{N_{\text{CV}}} c_i \text{Cov}(g_i, g_j) = -\text{Cov}(f, g_j). \tag{35}$$

If we let $A_j = -\text{Cov}(f, g_j)$ and $B_{ij} = \text{Cov}(g_i, g_j)$ then

$$\sum_{i=1}^{N_{\text{CV}}} B_{ij} c_i = A_j \qquad \text{or} \qquad \mathbf{c} = \mathbf{B}^{-1}\mathbf{A}. \tag{36}$$

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
