# Peer review of "Variance Reduction via Simultaneous Importance Sampling and Control Variates Techniques Using Vegas"

_SciPost Physics Codebases, doi:SciPost Phys. Codebases 28-r1.4 (2024) , SciPost Phys. Codebases 28 (2024)_

## Round 1 · Referee Report · Anonymous · 2023-10-17

Strengths

1. This paper is super well written and explain making him also available for non expert.

2. The author followed a list of example from another paper allowing to avoid bias in their presentation of the result.

3. The idea of the paper is original and important for the field of High Energy Physics due to the future High Luminosity run.

Weaknesses

1. The main result of the paper are not very exiting, in the sense that the gain over simply using VEGAS are often quite marginal.

2. The reason why the method is sometimes working and sometimes not is not investigated/explained in the paper.

3. The drawback/potential issue of control variates/importance sampling/VEGAS are not covered in the paper which can give a false impression to non expert user.

Report

This paper presents a code implementing an interesting way to combine two well known method of numerical integration. While the paper is well written, the results are not impressive. Consequently, I have some doubt that the method/code will be significantly used in the future.

In term of the acceptance criteria for SciPost Physics Codebases, all the criteria are passed with high color but the one for the "added value" which, is as said above, more border line (but certainly existing in some specific case). In conclusion my recommendation would be to publish this paper.

Given the weakness reported above, I'm suggesting below some modification/clarification to the paper in the hope to improve his clarity and impact. However I do not consider those changes as mandatory for publication.

Requested changes

As said in the above report, the requested changes are mainly suggestion to the authors on the point that I would have found interesting for them to comment.

1. Given the importance in our field of event generator code (Pythia, MadGraph, Sherpa, ...) on which many reader will think when reading this paper, it would be important to comment on the impact on event generation and in particular un-weighting efficiency.

2. Maybe it would be good to give more details on reference #10 and #11 in the introduction in order to provide a better picture on how novel your approach is compare to those two papers.

3. I have some doubt on the validity of Equation 19 if p(x) and g(x) are not normalised to one. My issue is to understand why the last term is within the integral and not outside of the integral. While not critical for the paper, I would kindly ask If he authors could double check that formula.

4. Maybe it would be interesting to comment on either section 2.3 or section 2.5 on the situation if the CV function is constant (which in section 2.5 is identical to the case when the grid is converging)

5. From the toy function used in the paper, I would claim that only two class of functions are providing significant result for the case with one CV, namely the Gaussian case and the Polynomial case. I would have like the authors to comment/investigate on the underlying reason for those absence (or presence) of such gain. My (maybe naive) guess is that this directly related to the fact that the function is (or not) separable dimension by dimension.

6. On table 1, the authors do not comment on the weird number on the RMS for the gaussian case with 8 dimension. In this case the RMS is significantly larger with CV, but according to table 2 the variance is reduced by 20%. Could the author check if they are a typo in one of the table or explained the reason of such apparent miss-match.

7. On Figure 2, it is not clear what "normalised variance" means in the sense that it is not clear if it is the normalised variance with or without CV for that iteration. The author can also consider to put both on the plot which can be instructive to understand such figure (see next point). This should also help to comment more on the reason about the correlation observed in that figure which are also not explained.

---

## Round 1 · Referee Report · Anonymous · 2023-12-14

Strengths

The paper is clearly written and presents a novel idea of using Vegas grids as Control Variates in high-dimensional Monte Carlo integration.

Weaknesses

The chosen application examples are rather academic and an actual real-live problem is missing.

Report

I am very sorry for the delay in reviewing the paper.

The paper is rather clearly written and presents the neat idea of employing Vegas grids as Control Variates when integrating high-dimensional functions by means of importance sampling based on Vegas. For a set of functions the authors illustrate the potential of the method to effectively reduce the variance of the MC integral estimate.

The obtained results indicate that in certain cases indeed a variance reduction is achieved, though for the price of an increase computational effort, that alternatively could have been invested in more samples in the standard Vegas integration. Nevertheless, I personally find it relevant and important to present the technique in a SciPost article.

However, prior to publication, the authors should address an additional aspect of their approach. Importance sampling through, e.g. Vegas, serves two purpose, (i) the direct evaluation of an integral estimate, (ii) the generation of event samples that follow the target distribution, being fully differential, possibly subject to an unweighting procedure.

While the authors clearly address the first case, they do not comment on how their method can be applied and possibly needs to be generalised when aiming for 'event generation'! According to Eq. (20) this seems straightforward, but I am wondering what happens when the phase space gets constrained through cuts, such that only a fraction of the generated events get accepted. Would this require reevaluation of $E_p$? The authors should discuss the applicability of their technique as a generative method.

Requested changes

1) The authors should consider adding more references at certain places, in particular concerning ML methods for integration in the introduction, where currently only Ref. [6] is cited. There are way more papers that should be quoted here.

2) Also in the introduction the authors state that the 'change of variables ... is difficult, if not impossible, to accomplish ... ' Its certainly difficult but this is precisely the target normalising flows or in general INN have been applied for recently and prove to be quite versatile. This should clearly be mentioned here.

3) Similarly, in the conclusion the authors should quote papers regarding the statements about inference techniques. Furthermore, just quoting Ref. [24] for using domain knowledge about the integrand is by no means adequate. This is a standard technique used in ALL event generators and not just MadGraph!

---

## Round 2 · Author Response

Our response and list of changes are shown under "List of changes".

---

## Round 2 · List of Changes

Referee 1

1) The authors should consider adding more references at certain places, in particular concerning ML methods for integration in the introduction, where currently only Ref. [6] is cited. There are way more papers that should be quoted here.

We thank the referee for the suggestion, and have updated the references in various places, including the introduction. As a result, the number of references has increased from 24 to 51.

2) Also in the introduction the authors state that the 'change of variables ... is difficult, if not impossible, to accomplish ... ' Its certainly difficult but this is precisely the target normalising flows or in general INN have been applied for recently and prove to be quite versatile. This should clearly be mentioned here.

We thank the referee for the comment. We actually agree with the referee and have completely rewritten that paragraph to avoid any misunderstanding.

3) Similarly, in the conclusion the authors should quote papers regarding the statements about inference techniques. Furthermore, just quoting Ref. [24] for using domain knowledge about the integrand is by no means adequate. This is a standard technique used in ALL event generators and not just MadGraph!

We thank the referee for the suggestion, and have updated the references.

Referee 2

As said in the above report, the requested changes are mainly suggestion to the authors on the point that I would have found interesting for them to comment.

  1. Given the importance in our field of event generator code (Pythia, MadGraph, Sherpa, ...) on which many reader will think when reading this paper, it would be important to comment on the impact on event generation and in particular un-weighting efficiency.

We thank the referee for the suggestion, we now added a comment in the second paragraph of the conclusions (the method is not meant to improve event generation).

  1. Maybe it would be good to give more details on reference #10 and #11 in the introduction in order to provide a better picture on how novel your approach is compare to those two papers.

Those references (25 and 26 in the revised version) are not targeting a high energy physics audience, and are primarily concerned with theoretical and mathematical aspects like convergence and asymptotic bounds. Our focus is on providing a practical tool which leverages the existing VEGAS infrastructure and can be used immediately by particle physicists (as well as other users of VEGAS).

  1. I have some doubt on the validity of Equation 19 if p(x) and g(x) are not normalised to one. My issue is to understand why the last term is within the integral and not outside of the integral. While not critical for the paper, I would kindly ask If he authors could double check that formula.

Equation (19) is correct. Note that being a probability distribution, p(x) is unit-normalized, as explicitly written in eqn (6). g(x) does not need to be unit-normalized for eqn (19) to be valid; it follows from the fact that the integral of g(x) equals the expected value of g(x)/p(x) for x sampled from p.

  1. Maybe it would be interesting to comment on either section 2.3 or section 2.5 on the situation if the CV function is constant (which in section 2.5 is identical to the case when the grid is converging)

A constant is not a good CV, since constants do not covary (Cov(f,constant)=0) and therefore there is no improvement per eqs. (17,18).

  1. From the toy function used in the paper, I would claim that only two class of functions are providing significant result for the case with one CV, namely the Gaussian case and the Polynomial case. I would have like the authors to comment/investigate on the underlying reason for those absence (or presence) of such gain. My (maybe naive) guess is that this directly related to the fact that the function is (or not) separable dimension by dimension.

We are actually not quite sure as to the underlying reason for the different performance gain observed in the toy experiments. Any comments from our side would be pure speculation.

  1. On table 1, the authors do not comment on the weird number on the RMS for the gaussian case with 8 dimension. In this case the RMS is significantly larger with CV, but according to table 2 the variance is reduced by 20%. Could the author check if they are a typo in one of the table or explained the reason of such apparent miss-match.

We thank the referee for their careful reading of the paper. We have double checked for typos. The two tables show errors that are computed differently. In Table 1 we compare the estimates to the true answer for the integral in order to compute the normalized rms error via eqn (31). In Table 2 we use the variance of the estimated values as the performance metric. It is possible for an estimator with lower variance to occasionally lead to a result which is farther away from the true value, simply due to statistical fluctuations. We think that this explains the rms error with CV being slightly larger than without CV for the case spotted by the referee.

  1. On Figure 2, it is not clear what "normalised variance" means in the sense that it is not clear if it is the normalised variance with or without CV for that iteration. The author can also consider to put both on the plot which can be instructive to understand such figure (see next point). This should also help to comment more on the reason about the correlation observed in that figure which are also not explained.

We thank the referee for the suggestion, we rewrote the figure caption of figure 2 and defined precisely what is meant by "normalized variance".

---

## Editorial Decision

published